# Promotion of Melanoma Cell Proliferation by Cyclic Straining through Regulatory Morphogenesis

**DOI:** 10.3390/ijms231911884

**Published:** 2022-10-06

**Authors:** Siyuan Huang, Zhu Chen, Xiaoqiang Hou, Kuankuan Han, Bingshe Xu, Miao Zhang, Shukai Ding, Yongtao Wang, Yingjun Yang

**Affiliations:** 1Materials Institute of Atomic and Molecular Science, Shaanxi University of Science and Technology, Xi’an 710026, China; 2School of Electro-Mechanical Engineering, Xidian University, Xi’an 710071, China; 3School of Medicine, Shanghai University, Shanghai 200444, China

**Keywords:** cyclic straining, melanoma, proliferation, mechanotransduction, cytoskeleton

## Abstract

The genotype and phenotype of acral melanoma are obviously different from UV-radiation-induced melanoma. Based on the clinical data, mechanical stimulation is believed to be a potential cause of acral melanoma. In this case, it is desirable to clarify the role of mechanical stimulation in the progression of acral melanoma. However, the pathological process of cyclic straining that stimulates acral melanoma is still unclear. In this study, the influence of cyclic straining on melanoma cell proliferation was analyzed by using a specifically designed cell culture system. In the results, cyclic straining could promote melanoma cell proliferation but was inefficient after the disruption of cytoskeleton organization. Therefore, the mechanotransduction mechanism of promoted proliferation was explored. Both myosin and actin polymerization were demonstrated to be related to cyclic straining and further influenced the morphogenesis of melanoma cells. Additionally, the activation of mechanosensing transcription factor YAP was related to regulatory morphogenesis. Furthermore, expression levels of melanoma-involved genes were regulated by cyclic straining and, finally, accelerated DNA synthesis. The results of this study will provide supplementary information for the understanding of acral melanoma.

## 1. Introduction

With high mortality and poor prognosis, melanoma is the most aggressive cutaneous cancer [1]. Clarified pathogenesis and progression mechanisms of melanoma are essential for the development of melanoma treatment. It is worth noting that the incidence and subtype of melanoma have obvious racial and regional differences [2,3]. Within Caucasian populations, the incidence of cutaneous and ocular melanomas is significantly higher than in dark-skinned individuals [4]. In addition, statistical data reveal that a higher incidence of melanoma is found on skin at fully exposed sites [5]. Based on this evidence, ultraviolet radiation is regarded by World Health Organization as a potential inducement for malignant melanoma, especially in fair-skinned populations. In this case, the pathological process of solar-radiation-induced melanoma has been well analyzed. UVA and UVB were able to damage DNA through the formation of reactive oxygen radicals [6] or the formation of forms of cyclobutene pyrimidine dimers and 6-4 photoproducts [7]. Additionally, oncogenic gene mutations, including BRAF, NRAS, and CDKN2A, were observed in most UV-induced melanomas [8]. On the other hand, in east Asia or in pigmented populations, the incidence of melanoma is significantly lower than in Europe and Oceania but has rapidly increased over the past decades [9]. The most prevailing subtype of melanoma in China, Korea, and Japan is acral melanoma [10]. Compared with solar-radiation-induced melanoma, primary lesions of acral melanoma are predominately located on the palms or soles of feet [11]. Unfortunately, research on acral melanoma is scarce. From common sense, doses of ultraviolet (UV) radiation in the acral region are significantly lower than on the skin at the dorsal back. In this case, acral melanoma is not believed to be caused by UV radiation [12]. Compared with UV-induced melanoma, acral melanoma has shown more genetic diversity [13] and no obvious predisposing genetic traits [14]. On the other hand, based on the clinical evidence of acral melanoma, the distribution of primary lesions at the palmoplantar surface has an obvious correlation with a mechanically stimulated region [15,16,17,18]. Additionally, high mechanical stress also correlates with oral mucosal melanoma [19]. Therefore, mechanical stimuli are considered another main etiology of melanoma. 

Over the past decades, the role of mechanical stimuli in the pathogenesis and therapeutic management of melanoma has gained extensive attention. Mechanical stimuli, including compression and tension, are always involved in daily human activities and simultaneously act on cells that are located at the palmoplantar surface. With the development of material engineering, these mechanical stimuli can be supplied by designed cell culture systems in vitro and have been demonstrated to be efficient in the regulation of various cell behaviors through mechanotransduction [20]. For instance, the stiffness of the substrate is an important factor in manipulating phenotype specification and the proliferation of melanoma cells [21]. In addition, the invasion process and nuclear envelope integrity of melanoma are affected by mechanical compression [22,23]. Previously, mechanotransduction has been demonstrated as the main pathway in cell-sensing extracellular biophysical stimuli [24]. At first, focal adhesions (FAs) were regarded as the main hub for cell–matrix interaction [25]. FAs are composed of various transmembrane proteins that directly bind to the extracellular matrix (ECM) and further regulate actin dynamics in adhesion structures and the contractility of stress fibers [26]. The activity of mechanosensing transcription factor Yes-associated protein (YAP) can be regulated by actin remolding through the Hippo signaling pathway [27]. Many studies have already revealed the critical role of YAP in mechanotransduction. As a downstream effector in the Hippo signaling pathway, YAP shuttles inside the nucleus and interacts with various transcription factors to activate specific genetic programs in sensing mechanical stimuli from the ECM [28]. In melanoma, the regulatory activity of YAP is able to regulate the phenotype differentiation of melanoma cells through YAP/PAX3/MITF or YAP/TEAD/SMAD transcription in sensing the stiffness of the ECM [29]. Therefore, it is necessary to elucidate the mechanism of morphogenesis and YAP-associated mechanotransduction in acral melanoma for sensing extracellular mechanical stimuli. Besides substrate stiffness and compression, dynamic straining also acts on the palmoplantar surface during daily activities. Unfortunately, even cyclic straining is critical in the regulation of other kinds of somatic or stem cell proliferation and differentiation [30,31], and their effects on melanoma progression are still not clear.

In this study, a cell culture system is designed and prepared to provide cyclic straining on melanoma cells in vitro. Then, the influence of cyclic straining on melanoma cell proliferation and DNA synthesis activity is characterized. Since FAs and the actin skeleton are essential in mechanotransduction, structures of the cytoskeleton and focal adhesions (FAs) are firstly characterized to elucidate the influence of cyclic straining on cellular morphogenesis. In addition, the activity of the transcription factor YAP (Yes-associated protein) and the expression levels of melanoma-related genes are analyzed to clarify the mechanism of cyclic straining promoting melanoma cell proliferation. Finally, the myosin inhibitor belbbistatin (Belbb.), actin polymerization cytochalasin D (Cyto D), and Rho-associated kinase (ROCK) inhibitor Y-27632 are applied to disrupt cytoskeleton organization and mechanotransduction.

## 2. Results and Discussion

### 2.1. Cell Culture System for Cyclic Straining

To provide cyclic straining on melanoma cells, a cell culture system was designed and prepared. As in the illustration of the cell culture system shown in Figure 1A, melanoma cells were seeded on flexible PDMS film. With a pressure change in the bottom cavity, the PDMS film presented cyclic concave and convex deformation (Figure 1B and Appendix A). Indeed, the cyclic straining on melanoma cells was provided by the tensile deformation of PDMS. Eight units were designed and prepared in the PTFE panel for parallel experiments. In addition, the bottom cavity of each unit was connected to a double-channel syringe pump to obtain the pressure change. After melanoma cells were treated with cyclic straining and inhibitors for 24 h, their cell viability was characterized. As shown in Figure 1B, cyclic straining, belbbistatin, Y-27632, and cytochalasin D had no significant influence on cell viability and were suitable for further experiments. 

### 2.2. Melanoma Cell Proliferation

After melanoma cells were stimulated for 72 h, the proliferation of melanoma cells was assessed. In the results, the number of mechanically stimulated (MS) melanoma cells was higher than the cells without mechanical stimuli (Figure 2B). Based on previous evidence, extracellular biophysical stimuli are converted into electrochemical signals through mechanotransduction by regulating cytoskeleton-associated morphogenesis [32]. In this case, both myosin and actin polymerization were inhibited to explore the role of cytoskeleton organization in the mechanosensing of cyclic straining. As shown in Figure 2B, with the treatments of myosin inhibitor blebbistatin and actin polymerization inhibitor cytochalasin D, the amount of proliferated melanoma cells was significantly decreased (Figure 2B). In addition, in a comprehensive analysis of the inhibitor-treated proliferation results without mechanical stimulation, the positive effect of cyclic straining disappeared after myosin and actin polymerization inhibitor treatments (Appendix A). Consequently, the cytoskeleton, including myosin and actin filaments, was indicated as a critical regulator in sensing cyclic straining. Furthermore, the critical role of the ROCK signaling pathway in the regulation of actomyosin contraction and stress fiber formation has been revealed by previous research [33]. Therefore, Y-27632 was also applied to investigate the role of the ROCK signaling pathway in sensing cyclic straining. As a result, after the treatment of Y-27632, the amount of proliferated melanoma cells was increased but independent of mechanical stimuli (Figure 2B). According to a previous report, Y-27632 was able to promote BRAF-mutated melanoma cell proliferation through the activation of AKT and ERK [34]. Additionally, the enhanced proliferation ability of melanoma cells after Y-27632 treatment was independent of mechanical stimuli. These results indicated that Y-27632-activated proliferation was independent of mechanotransduction in sensing cyclic straining. 

In summary, the proliferation ability of melanoma cells was promoted by cyclic straining but inhibited after the disruption of cytoskeleton organization.

### 2.3. DNA Synthesis Activity

As shown in Appendix A, with the treatment of Cyto D, the number of melanoma cells was slightly decreased after mechanical stimulation. It was considered the result of cell detachment during the cyclic deformation of PDMS film after the inhibition of actin filaments polymerization. Thus, in order to precisely evaluate the influence of cyclic straining on melanoma cell proliferation, DNA synthesis activity was also explored by EdU assay. Since new DNA is always synthesizing during the mitosis of cells, DNA synthesis activity is able to reveal the potential for cell proliferation. The results of EdU incorporation showed a similar tendency to the CCK-8 results (Figure 3 and Figure 4). The DNA synthesis activity was significantly enhanced by cyclic straining and inhibited by disrupted myosin and actin filament formation. On the other hand, DNA synthesis activity was enhanced by the Y-27632 treatment. This result was consistent with the results of CCK-8. Consequently, it was another piece of evidence to support regulated myosin, and actin polymerization was the principal factor in cyclic straining promoting melanoma proliferation. It was interesting that with the treatment of cyclic straining, the positive effect on A375 cells was greater than on B16 cells. In a previous report, after the interference of mechanosensing ion channel Piezo 1, the proliferation ability of melanoma cells was inhibited [35]. Specifically, the decreased positive ratio of Ki67 in A375 cells was greater than the decreased value in B16 cells. This means the A375 cells are more sensitive to Piezo 1 than B16 cells. As a mechanosensing ion channel, Piezo 1 is able to allow the permeation of mechanosensitive Ca^2+^ during mechanical stimulation [36]. There is also another piece of evidence to show that the activated Piezo 1 channel is able to promote cancer cell proliferation through the AKT/mTOR pathway [37]. Therefore, regulated Piezo 1 activity is another potential factor in the regulation of melanoma cell proliferation.

In summary, DNA synthesis activity was also enhanced by cyclic straining and inhibited after the disruption of cytoskeleton organization. In addition, the regulated DNA synthesis activity was independent of the ROCK signaling pathway. Furthermore, mechanosensing ion channel Piezo 1 was also indicated as another regulator in sensing cyclic straining.

### 2.4. Cell Morphology and FAs

Based on previous reports, myosin activity and actin polymerization were demonstrated to be tightly related to cellular morphogenesis [38,39]. Furthermore, numerous studies have revealed that cell morphogenesis is critical in mechanical-stimuli-regulated biological processes [40,41,42,43]. Therefore, the cell morphology and cytoskeleton structure were characterized to elucidate the internal mechanism of cyclic straining effects on melanoma cell proliferation. As shown in Figure 5 and Figure 6, cell spreading and elongation were facilitated by mechanical stimulation in both A375 cells and B16 cells. In addition, with the disruption of myosin and actin filaments, the spreading area and aspect ratio of mechanically stimulated melanoma cells were significantly decreased. These results reveal cell spreading and elongation are regulated by cyclic straining through the rearrangement of the cytoskeleton. In our previous work and other research, well-spread or elongated cell morphology was able to accelerate nuclear DNA synthesis and further promote cell proliferation [44,45,46]. Consequently, well-spread and elongated cell morphology was considered an efficient factor in the regulation of melanoma cell proliferation through cyclic straining. There was another interesting phenomenon: for B16 cells, Y-27632 treatment was able to promote melanoma cell proliferation while simultaneously inhibiting cell spreading and elongation (Figure 5). For A375 cells, Y-27632 could inhibit cell elongation but also accelerate cell proliferation (Figure 6). These phenomena reveal that the Y-27632-promoted melanoma cell proliferation was independent of cyclic-straining-induced regulatory morphogenesis.

Additionally, the structure of FAs was also demonstrated to be related to DNA synthesis activity by a previous report [47]. Thus, the effects of cyclic straining on FAs were also investigated. In this study, vinculin was stained to indicate cytoplasm FA complexes. As shown in Figure 5 and Figure 6, cyclic straining was beneficial for melanoma cells to gain more FAs but inefficient in the regulation of the FAs’ size. Compared with the result of cell spreading, the larger spreading area was always associated with more FAs. With the inhibition of actin polymerization, the spreading area of melanoma cells was significantly decreased. In the results, with the treatment of myosin inhibitor, the decreased total FAs were also associated with a decreased spreading area. These results indicate that the cyclic-strained melanoma cells with higher DNA synthesis activity were always well spread and, further, generated more FAs. These results are consistent with a previous report that more FAs had positive effects on the acceleration of nuclear DNA synthesis [48]. Thus, the regulatory morphogenesis of melanoma cells with well-spread morphology and more FAs is considered an efficient pathway in the regulation of melanoma cell proliferation. 

In summary, cell spreading and elongation were facilitated by cyclic straining. Additionally, with an increased spreading area, the total area of FAs was also increased after mechanical stimulation treatment. Based on previous reports, the cyclic-straining-induced cell spreading and elongation were beneficial for the facilitation of cell proliferation. 

### 2.5. YAP Activity and Melanoma Gene Expression

As a crucial mechanosensory and mechanotransducer, Yes-associated protein (YAP) was also characterized in this study [49]. Since the transcription factor YAP is only functionalized within the karyotheca to regulate target gene expression, the distribution of YAP was analyzed to indicate YAP activity. As shown in Figure 7A,B, with mechanical stimulation, YAP transcription factors were predominately transported into nuclei. In contrast, more YAP was silenced within the cytoplasm of melanoma cells without cyclic straining. With the inhibition of actin polymerization, cells were significantly contracted, and it was difficult to distinguish the nuclear region from the cytoplasm in A375 cells. In this case, a myosin inhibitor was also applied to investigate the role of the cytoskeleton in the activation of YAP. In the results, with disrupted myosin and actin filaments, YAP was not able to be transported into nuclei after mechanical stimulation. These results revealed that YAP was activated by cyclic straining through the regulation of cytoskeleton-organization-induced morphogenesis. In addition, with the Y-27632 treatment, the localization of YAP was not significantly changed. This means the ROCK pathway was inefficient in the regulation of YAP activation. 

In the YAP signaling pathway, cell behavior variation has always been regarded as the result of regulatory gene expression. Moreover, the malignancy of melanoma cells is typically accompanied by recognized gene mutations [50]. According to this basis, the expression level of melanoma- and mechanotransduction-involved genes was analyzed in B16 cells. In the results, mechanical stimulation promoted NRAS and BRAF expression but inhibited NF1 expression (Figure 7C). In particular, as a typical melanoma oncogene, BRAF was able to accelerate melanoma cell proliferation through the lineage-specific factor MITF [51]. The NRAS mutation is also efficient in regulating melanoma proliferation [52]. On the other hand, as a tumor suppressor, the loss of NF1 commonly occurs in cutaneous melanoma [53]. All these regulatory gene expressions in mechanically stimulated melanoma cells are associated with the malignancy of melanoma and may be regulated by mechanosensing transcription factors. In this case, the expression level of mechanotransduction-associated genes was also characterized. As shown in Figure 7C, cyclic straining only affected YAP location but not the expression of YAP/TAZ. However, the expression level of YAP/TAZ co-transcription factors (TEAD1 and TEAD2) and mechanoregulators (SMAD1 and SMAD2) was impacted by cyclic straining or cytoskeleton disruption. Thus, the regulatory expression of melanoma-specific genes may relate to the affected transcription factors through mechanotransduction and, finally, impact DNA synthesis activity through regulatory morphogenesis. In addition, since the mechanosensing ion channel Piezo 1 was also considered a regulator in sensing cyclic straining, the expression level of Piezo 1 and Piezo 2 was also characterized. In the results, the expression level of Piezo ion channels was independent of mechanical stimulation or inhibitor treatment. This means the cyclic straining may only switched the Piezo channels instead of regulating their expression.

In summary, the localization of YAP and the expression of melanoma-associated mutated genes were regulated by cyclic straining and cytoskeleton inhibitors. The results of YAP positioning demonstrated that the YAP-associated mechanotransduction pathway was activated during the cyclic straining treatment. In addition, the regulatory gene expression also indicated the potential relationship between the mechanotransduction pathway and melanoma progression.

## 3. Materials and Methods

### 3.1. Design and Preparation of Cell Culture System

A cell culture system with a polytetrafluoroethylene (PTFE) mold and flexible polydimethylsiloxane (PDMS) film was designed and prepared to provide cyclic straining on melanoma cells. As shown in Figure 1A, the PTFE mold contains the PDMS film in a separate cavity. The cavity in the upper PTFE panel, with 12 mm inner diameter and 10 mm depth, was designed for cell culture. In addition, the cavity at the bottom PTFE panel, with 12 mm inner diameter and 10 mm depth, was connected with a syringe pump (LSP02-2B, Longer Precision Pump Co., Ltd., Baoding, China) to obtain adjustable air pressure within the lower cavity. The PDMS film with 100 μm in thickness was commercially purchased from Hangzhou Bald Advanced Materials Technology Co., Ltd. and firstly cut into a circular shape of 14 mm diameter. After oxygen plasma treatment (100 sccm, 40 W, 180 s, VP-S5, Guangzhou SunJune Technolog Co., Ltd., Guangzhou, China), the PDMS film was placed between the upper and bottom cavities of the PTFE mold for bottom cavity sealing and cell adhesion. With the reciprocating motion of the syringe pump, the positive and negative air pressures were alternately applied within the sealed bottom cavity, and this drove the cyclic tensile deformation of the PDMS film. Before cell culture, the cell culture system was sterilized with 70% ethanol for 20 min and rinsed with PBS solution twice.

### 3.2. Cell Culture

Mouse melanoma cells (B16) and human melanoma cells (A375) were purchased from Procell Lifer Science & Technology Co., Ltd. (Wuhan, China) and subcultured in DMEM medium (Mishu (Xi’an) Biotechnology Co., Ltd., Xi’an, China) supplied with 10% FBS (Biological Industries Israel Beit Haemek Ltd., Beit Haemek, Israel) and 1% penicillin–streptomycin (Mishu (Xi’an) Biotechnology Co., Ltd., Xi’an, China); 1 mL cell suspension (5000 cells/mL) was seeded in each well of the cell culture system. After being cultured in a humidified CO_2_ incubator for 24 h, the medium with suspended cells was refreshed, and the syringe pump was operated to provide cyclic straining for the melanoma cells. The injection volume and speed of the syringe pump were set as 1 mL and 1 mL/s to obtain 1 Hz tensile deformation of PDMS film. Melanoma cells were stimulated for 6, 24, or 72 h in different experiments.

### 3.3. Belbbistatin, Cytochalasin D and Y-27632 Treatment

The myosin inhibitor belbbistatin (Shanghai Aladdin Biochemical Technology Co., Ltd., Shanghai, China), actin polymerization inhibitor cytochalasin D (Shanghai Aladdin Biochemical Technology Co., Ltd., Shanghai, China) and ROCK pathway inhibitor Y-27632 (Shanghai Aladdin Biochemical Technology Co., Ltd., Shanghai, China) were applied to disrupt cytoskeleton organization. After cell culture in the cell culture system for 24 h, the medium with suspended cells was refreshed by cell-cultured medium with 100 ng/mL blebbistatin or 100 ng/mL cytochalasin D or 2 μM Y-27632. Then, cyclic straining was applied for a further 6, 24, and 72 h for each experiment.

### 3.4. Cell Viability Analysis

Cells were stained by calcein-AM/PI assay (Beijing Solarbio Science & Technology Co., Ltd., Beijing, China) to investigate the influence of cyclic straining and inhibitors on cell viability. After cell culture in the cell culture system and being cyclic-strained for 6 h, samples were firstly rinsed with prewarm PBS and stained with calcein-AM/PI according to the product manual. Fluorescent images of the stained sample were recorded by a fluorescence microscope (MF52-N, Guangzhou Micro-shot Technology Co., Ltd., Guangzhou, China). 

### 3.5. Actin, Vinculin, and Nuclei Staining

The actin of melanoma cells was stained to calculate the cell spreading area and aspect ratio. Simultaneously, vinculin was stained to analyze the structure of FAs in the cytoplasm. After melanoma cells were mechanically stimulated for 6 h, the samples were firstly fixed by 4% cold paraformaldehyde for 10 min and subsequently treated by 1% Triton X-100 and 0.02% Tween-20 at room temperature for 30 min. Then, the samples were blocked with 1% *w*/*w* bovine serum albumin (BSA, Shanghai Aladdin Biochemical Technology Co., Ltd., Shanghai, China) at room temperature for 30 min. After the blocking process, an aqueous solution of rabbit anti-vinculin antibody (Abcam plc.) at a dilution ratio of 1:100 in 2% BSA was incubated with samples at 37 °C for 1 h. The samples were rinsed with 0.02% Tween-20 aqueous solution at room temperature for 30 min. After being washed with PBS three times, the samples were incubated in an aqueous solution of Alexa Fluor-488-labeled donkey anti-rabbit IgG antibody (Invitrogen, Carlsbad, CA, USA) at a dilution ratio of 1:1000 at 37 °C for 1 h. Finally, actin filaments were stained by incubating the samples with Alexa Fluor-594 phalloidin (Beijing Solarbio Science & Technology Co., Ltd., Beijing, China) at a dilution ratio of 1:40 in PBS at room temperature for 20 min. Nuclei were stained with 10 mg/mL of Hoechst 33,258 (Beijing Solarbio Science & Technology Co., Ltd., Beijing, China). After being rinsed with PBS, the samples were observed and recorded using a fluorescence microscope. The total amount and average size of focal adhesion were calculated using ImageJ software according to a previous report [54]. The contrast of fluorescent images was adjusted to elucidate stained FAs. The limitations of circularity and area were set during the calculation of FAs to avoid the nonspecific absorption of fluorochrome in cytoplasm and nuclei. The spreading area and aspect ratio of cells were calculated from actin-stained images using ImageJ software. More than 30 cells from 3 independent experiments were analyzed.

### 3.6. CCK-8 and EdU Incorporation Assay

The proliferation of melanoma cells was characterized by CCK-8 and EdU incorporation assay. After being cell-cultured and stimulated for 72 h, the medium was replaced by CCK-8 reagent (1:10 diluted in cell culture medium, Mishu (Xi’an) Biotechnology Co., Ltd., Xi’an, China) and further incubated for 1 h in a humidified incubator. The absorbance at 450 nm was recorded. Three parallel experiments were analyzed to get quantitative data. Before the EdU incorporation assessment, melanoma cells were firstly cultured in FBS free medium for 8 h before cell culture. After being cell-cultured in the system for 24 h and stimulated for another 24 h, a BeyoClick^TM^ EdU cell proliferation kit with Alexa Fluor 488 (Beyotime Biotechnology, Haimen, China) was applied according to the product manual. The stained samples were observed and recorded by a fluorescent microscope. More than 50 cells from 3 independent experiments were analyzed to get quantitative data.

### 3.7. YAP and Nuclei Staining

After melanoma cells were stimulated by cyclic straining for 24 h, the YAP and nuclei were stained to analyze YAP localization. Firstly, the samples were fixed with 4% cold paraformaldehyde for 10 min and permeabilized with 1% Triton X-100 for 2 min. Then, the samples were blocked with 2% BSA and incubated with 1:100 diluted rabbit anti-YAP antibody (Proteintech Group, Inc., Rosemont, IL, USA) for 1 h in a humidified incubator. After being rinsed with PBS, the samples were subsequently incubated with Alexa Fluor-488-labeled donkey anti-rabbit IgG antibody (Abcam plc.) at a dilution ratio of 1:1000 for another 1 h. Finally, the nuclei were stained by Hoechst 33,258. The nuclear or cytoplasmic localization of YAP was analyzed according to a previous article [48]. More than 50 cells from 3 independent experiments were analyzed to get quantitative data.

### 3.8. Eukaryotic Transcriptome Sequencing and Differentially Gene Expression Analyses

After melanoma cells were cultured in the system for 24 h and mechanically stimulated for 72 h, RNA was extracted and purified by using TRIzol^®^ reagent (Thermo Fisher Scientific, Inc., Waltham, MA, USA). Then, the Illumina transcriptome sequencing was performed to analyze the differential gene expression. The normalized FKPM value was analyzed to generate a gene expression heatmap.

### 3.9. Statistical Analysis

The significant difference among samples was calculated using a one-way analysis of variance (ANOVA) with Tukey’s post hoc test for multiple comparisons. The data are presented as means ± standard deviations (SDs). It is considered to be a statistically significant difference when *p* < 0.05.

## 4. Conclusions

The effects of cyclic straining on melanoma cell proliferation through mechanotransduction were systemically analyzed in this study. As shown in Figure 8, the possible mechanism of promoted cell proliferation is explained by the following pathway. Firstly, myosin and actin polymerization are regulated by cyclic straining and further influence cell spreading and elongation. Then, the regulatory morphogenesis of melanoma cells induces the activation of mechanosensing transcription factor YAP. The activated YAP and associated transcription factors may be targeted on the melanoma-involved genes and, finally, accelerate DNA synthesis activity. These results provide supplementary information for understanding the pathological process of mechanical-stimulation-induced melanoma.

## Figures and Tables

**Figure 1 ijms-23-11884-f001:**
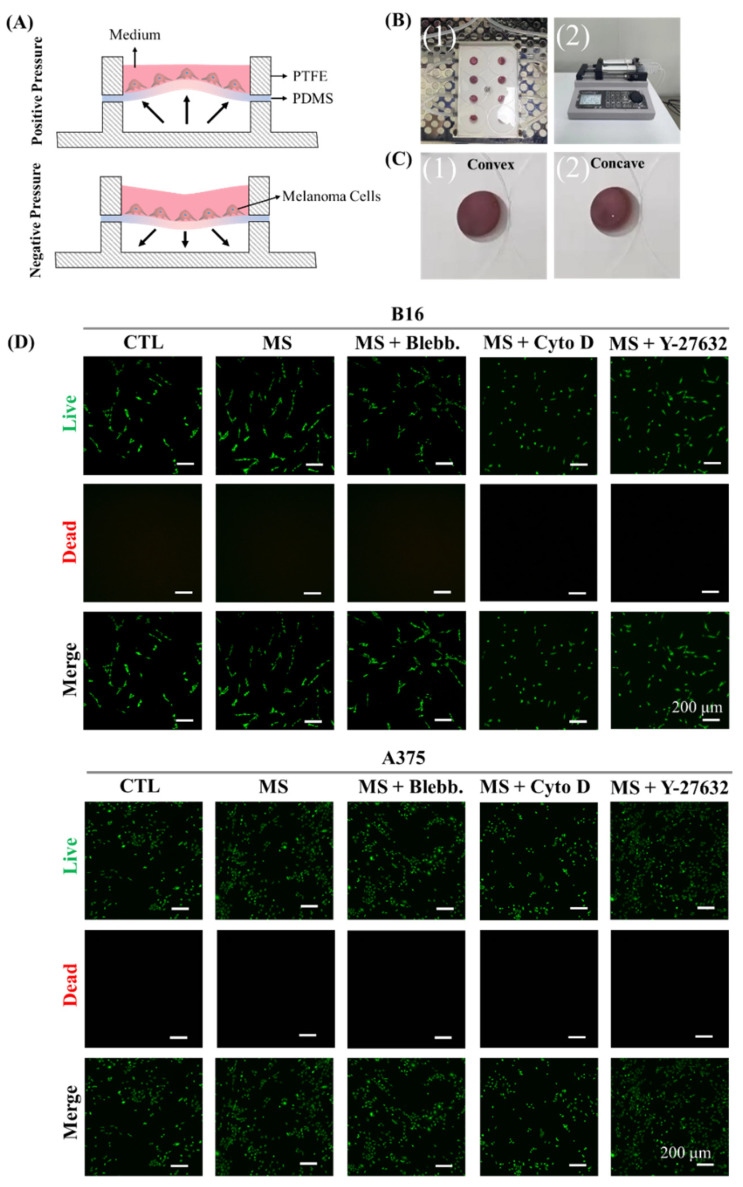
Cyclic-strained cell culture system and cell viability. (**A**) Illustration of the cell culture system. (1) PTFE panel with 8 cell culture wells. (2) Syringe pump. (**B**) Convex (1) and concave (2) deformation of PDMS film. (**C**) Representative fluorescent images of live/dead staining. CTL: control. MS: mechanically stimulated. MS + Blebb.: mechanically stimulated with Blebb treatment. (**D**) MS + Cyto: mechanically stimulated with Cyto D treatment. MS + Y-27632: mechanically stimulated with Y-27632 treatment.

**Figure 2 ijms-23-11884-f002:**
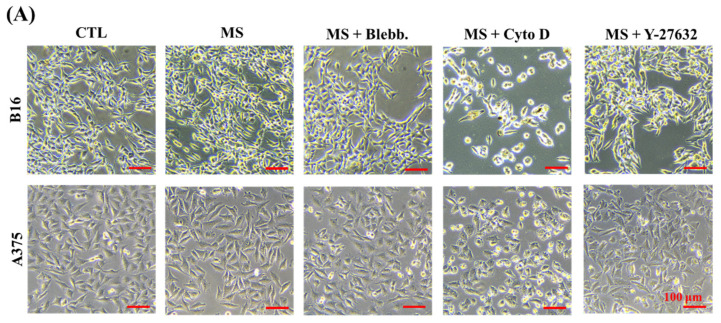
Influence of cyclic straining and inhibitors on melanoma cell proliferation. (**A**) Representative images of melanoma cells with cyclic straining and inhibitor treatment. (**B**) Quantitative data of melanoma cell proliferation from CCK-8 assay. Data are presented as means ± SDs (*n* = 3). ** *p* < 0.01, *** *p* < 0.001, N.S.: Not Significant.

**Figure 3 ijms-23-11884-f003:**
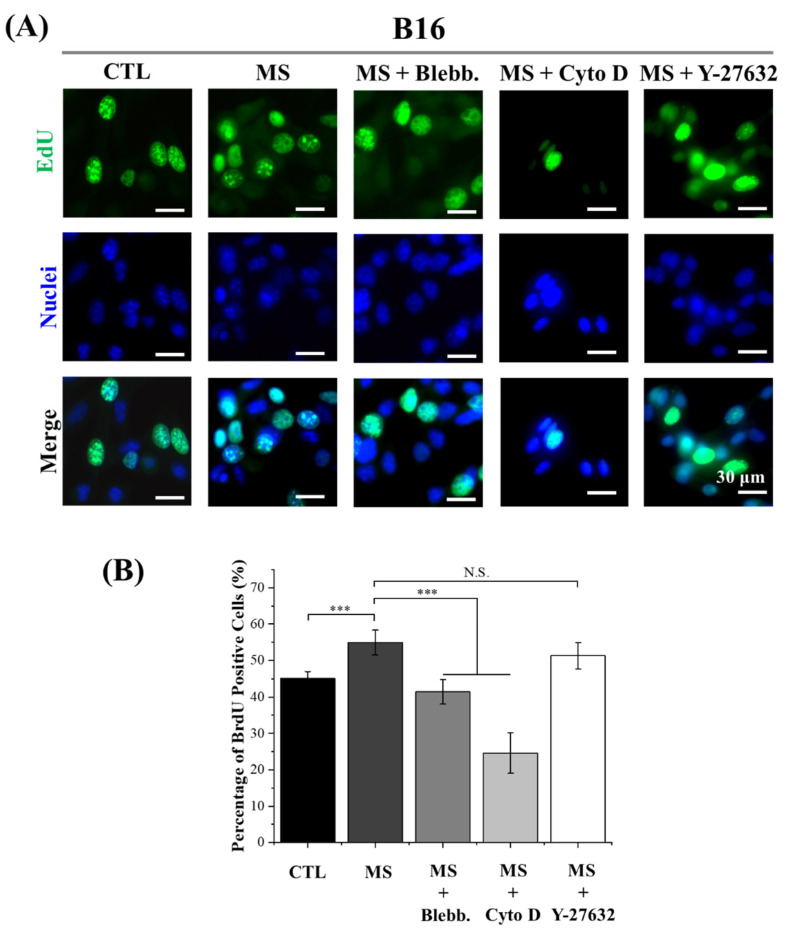
Influence of cyclic straining and inhibitors on the DNA synthesis activity of B16 cells. (**A**) Representative fluorescent images of EdU incorporation. (**B**) Quantitative data of EdU incorporation assay. Data are presented as means ± SDs (*n* = 3). *** *p* < 0.001, N.S.: Not Significant.

**Figure 4 ijms-23-11884-f004:**
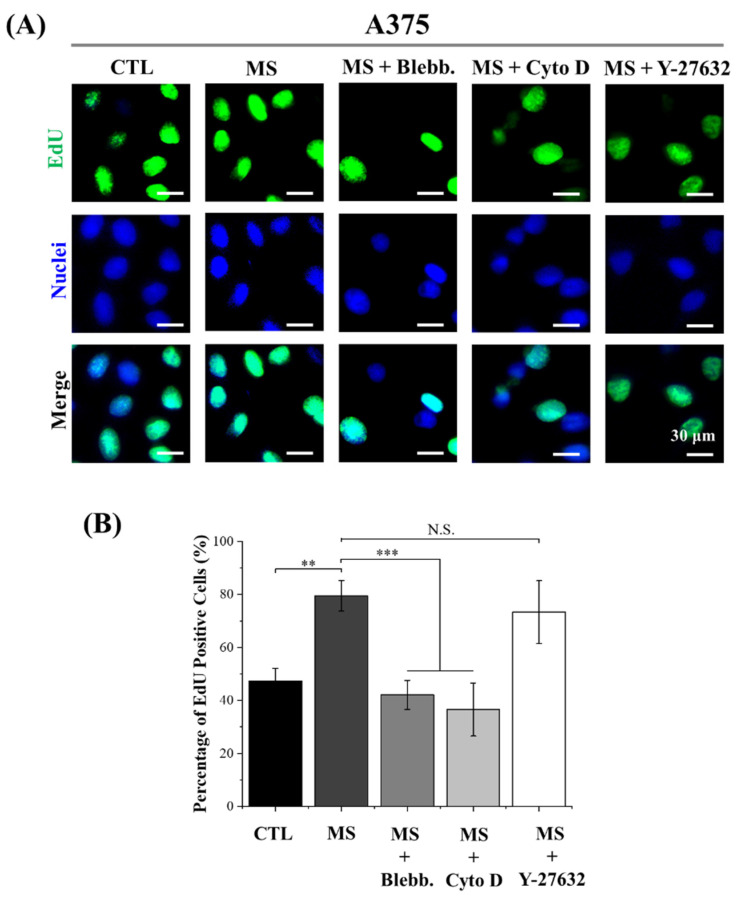
Influence of cyclic straining and inhibitors on the DNA synthesis activity of A375 cells. (**A**) Representative fluorescent images of EdU incorporation. (**B**) Quantitative data of EdU incorporation assay. Data are presented as means ± SDs (*n* = 3). *** *p* < 0.001, ** *p* < 0.01, N.S.: Not Significant.

**Figure 5 ijms-23-11884-f005:**
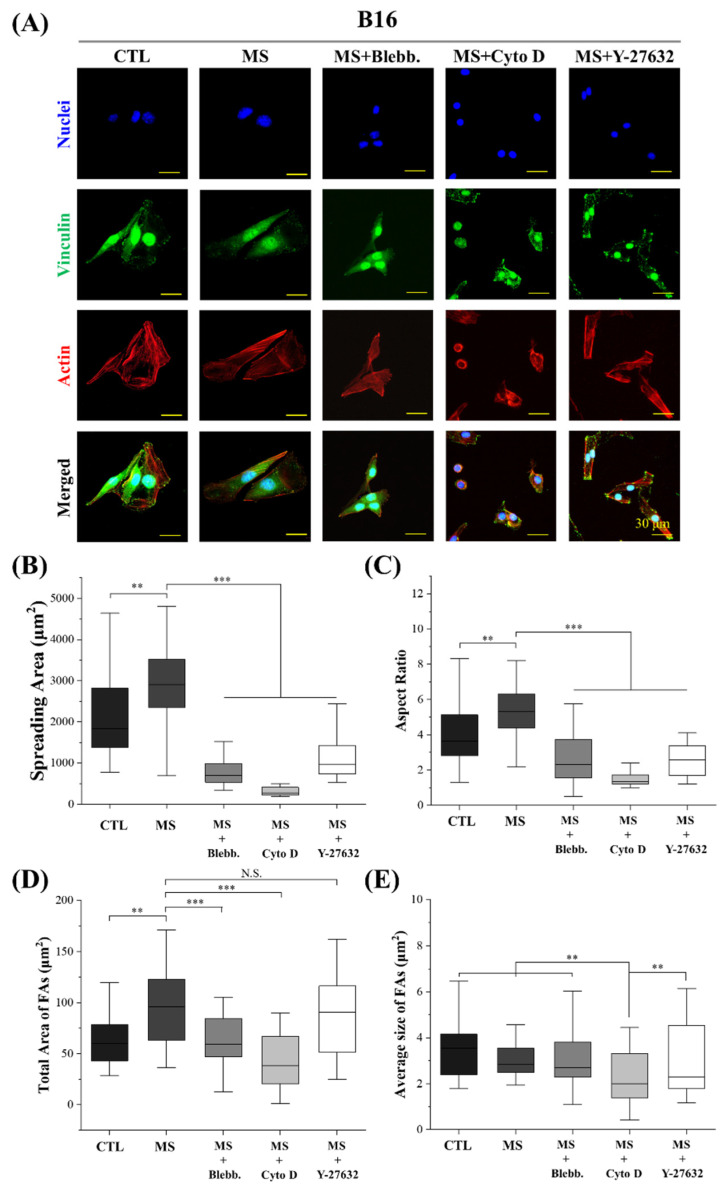
Influence of cyclic straining and inhibitors on the morphogenesis of B16 cells. (**A**) Representative fluorescent images of actin, vinculin, and nuclei-stained B16 cells. (**B**) Spreading area of B16 cells (*n* > 30). (**C**) Aspect ratio of B16 cells (*n* > 30). (**D**) Total size of focal adhesions in B16 cells. (*n* > 30). (**E**) Average size of focal adhesions in B16 cells (*n* > 30). ** *p* < 0.01, *** *p* < 0.001, N.S.: Not Significant.

**Figure 6 ijms-23-11884-f006:**
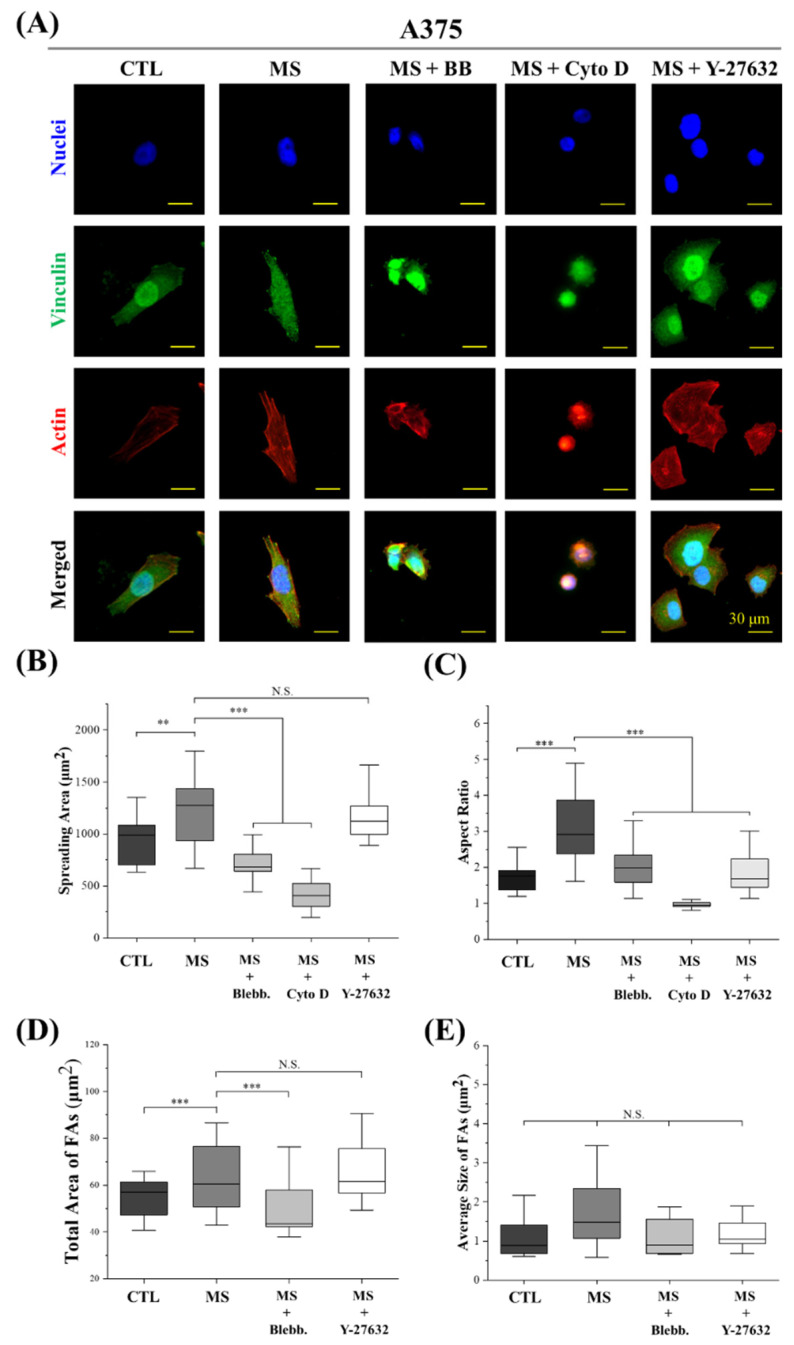
Influence of cyclic straining and inhibitors on the morphogenesis of A375 cells. (**A**) Representative fluorescent images of actin, vinculin, and nuclei-stained A375 cells. (**B**) Spreading area of B16 cells (*n* > 30). (**C**) Aspect ratio of A375 cells (*n* > 30). (**D**) Total size of focal adhesions in A375 cells. (*n* > 30). (**E**) Average size of focal adhesions in B16 cells (*n* > 30). ** *p* < 0.01, *** *p* < 0.001, N.S.: Not Significant.

**Figure 7 ijms-23-11884-f007:**
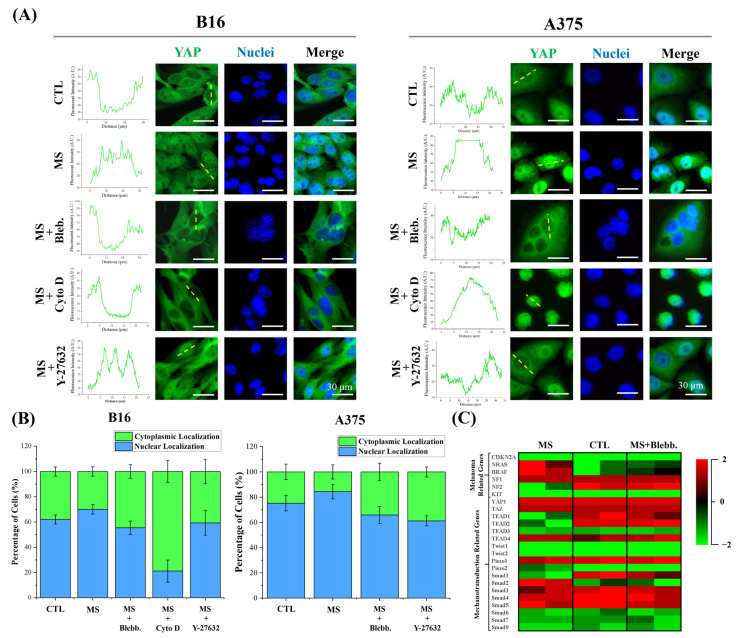
Influence of cyclic straining and inhibitors on YAP localization and gene expression. (**A**) Representative fluorescent images of YAP and nuclei-stained melanoma cells. (**B**) Quantitative data for YAP localization (*n* = 3). (**C**) Heat map of RNA sequencing transcriptome analysis for selected genes. Data are presented as means ± SDs.

**Figure 8 ijms-23-11884-f008:**
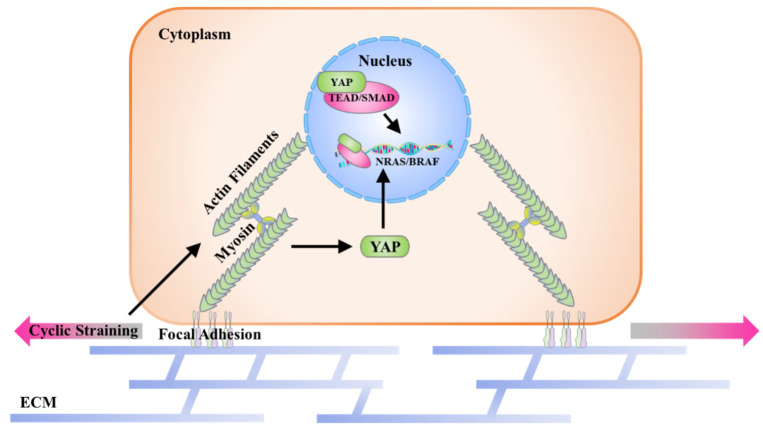
Schematic representation of the potential mechanism in cyclic-straining-promoted proliferation through regulatory morphogenesis.

## Data Availability

The data presented in this study are available upon request from the corresponding author. The data of the eukaryotic transcriptome sequencing were deposited in the Gene Expression Omnibus (GEO) with the accession code GSE214638.

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
