# Peer review of "Promotion of Melanoma Cell Proliferation by Cyclic Straining through Regulatory Morphogenesis"

_ijms, 2022, doi:10.3390/ijms231911884_

Round 1
Reviewer 1 Report
Comments to the authors:
Dear authors of the manuscript ID:IJMS-1872703,
please find bellow my comments to the manuscript. In general terms I found the work presented with results not enough to corroborate your conclusions and many assumptions based on other’s work are made, without really going deep and accurate to the results.
I have found your introduction a bit superficial, with some scientific language that can and should be more objective and precise.
For example, In line 30, the authors should be more precise since the dorsal back skin is the most common location for Caucasian cutaneous melanomas for men but not in female, that is in the legs.
Also in the introduction that authors could make perhaps a more precise overview of the mechanotransduction and signalling impact on melanoma since it is mentioned “over recent years” in line 45 but the references used are back to the year 2006; not so recent years.
In my opinion figure 1 should be presented when results start and not before as it is the case.
In figure 2, more accurate information should be presented, like illustrating what means MS,
ctl and MS+Blebb. Also, in 2C and D the live/dead assay used by the authors should be presented in a better way. All the images in red of Dead cells are devoid of fluorescence, there is no point of including them in the figure and explain that cell viability in figure 2D was obtained by the ratio Live/Dead cells?
I do not understand how the authors measured here live/dead when dead cells are absent in all the conditions, including in Blebb. Treatment.
The authors mention throughout the text of the manuscript melanoma cells when in fact the only cell line used in this study was the B16 mouse melanoma cell line. This murine cell line as been vastly used to metastasis. The attempt to correlate these studies using a mouse melanoma cell line with human acral melanomas mentioned in the introduction is again to speculative and far from what can be obtained with Human cell lines or cells obtained from human melanomas.
In line 91; only Blebbistatin was used, inhibiting the functioning of actin cytoskeleton. But this is not the only drug capable of inhibiting actin cystokeleton; a more direct disruption of actin cytoskeleton could be accomplished by using cytochalasin D for example that impairs filamentous actin. Additionally, Cell’s cytoskeleton in not only composed by actin; importantly microtubular cytoskeleton plays important roles in cells ability to proliferate and on nuclear morphogenesis. This way the authors should provide additional and more exhaustive studies of disturbing cell’s cytoskeleton and assess the proliferation capacity and nuclear integrity and DNA synthesis.
Figure 3 is showing vinculin, a focal adhesion protein in the nucleus. Vinculin is a cytoplasm attic actin-binding protein enriched in focal adhesions and adherens junctions. Not sure if we should see vinculin staining in B16 nucleus as the authors present. Also, the actin cytoskeleton assessment was done using phalloidin, a phallotoxin that should show only actin and not a fluorescence in cell cytoplasm not resembling actin cytoskeleton. Probably, the SOP used for the staining presented in figure 3 might be improved in order to show more accurately, the cell’s actin cytoskeleton and not detecting vinculin in the nucleus of the cells.
In line 115 the correlation made by the authors with more DNA nuclear synthesis in not obvious. Then, the conclusion make by the authors here in my opinion is not corroborated by the results presented and is too speculative.
In line 132 and forward the authors try to correlate the transcriptome of the most common mutated genes in cutaneous melanoma (BRAF and NRAS) with proliferation that is one of the characteristics/functions of such genes, like MITF here mentioned. Again in my opinion this is to speculative and extrapolated, since MITF is a master regulator of melanocyte survival. Besides proliferation MITF controls several other functions on melanocytes, like melanocyte apoptosis, melanogenesis, regulates ECM genes important for invasion of melanoma cells, etc…
Thus, I believe the authors should not use this as an example of the effect on proliferation of these particular melanoma cells.
The conclusions taken by the authors are not supported by the results that were provided. To much speculative the interpretation of the data.
In Conclusion, I believe the data presented in the manuscript is still far from being accepted for publication and is my advice to the authors to improve the data before a resubmission.
Author Response
Thank you for your comments. Please see the attachment for responses.

Reviewer 2 Report
The authors analyzed the role of mechanical stimuli in the pathogenesis of melanoma. They designed a cell culture system to provide cyclic straining on melanoma cells in vitro. They showed that cyclic straining had no significant influence on cell viability but stimulated melanoma cell proliferation and DNA replication. Cell spreading, cell elongation and focal adhesions were increased by mechanical stimulation. They further showed that, after cyclic straining, the yes-associated proteins (YAP) were predominately found into nuclei, a sign of its activity, suggesting that YAP could act as a mechanosensor and a mechanotransducer in this study.
The study of mechanical stimuli on melanoma is very interesting in particular in the context of acral melanoma. However, the results are based on only one melanoma cell line, the B16 mouse melanoma cell line. Experiments should be repeated on other cell lines and in particular human melanoma cells. The B16 cell line is not very representative of human melanoma showing none of the genetic alterations described in human tumors. Cell with different genetic alterations could be used and in particular melanoma cell lines mutated on KIT as KIT alterations are specifically found in acral and mucosal melanoma.
Figures 1 to 3: The results of the experiments using control cells treated with belbbistatin are not shown.
Figure 4: To confirm YAP activation, the authors should look at the expression of YAP targets genes in their RNA sequencing transcriptome analysis.
Author Response

(The authors gave the same response as above.)

Round 2
Reviewer 1 Report
After a carefull reading of the second version of the manuscript I still have major concerns about the study.
Although the authors claim that a revision in the introduction was made, I still find it superficial; with a lot of “obvious” marks made by the authors but with no innovative and careful revision of the literature concerning melanoma, (It is true changes were made relative to this but still superficial. If in men cutaneous melanomas are mostly found in the dorsal back/head, not only men develop cutaneous melanomas; in women this melanoma subtype is more frequently detected in the legs). A more accurate overview of the mechanotransduction cell system is needed to better understand the rationale of cytoskeleton disruption, YAP activity, etc…
In summary, I believe the introduction of the manuscript can be greatly improved.
Relative to the in vitro cell culture system presented in figure 1, I believe controls to differentiate this way of creating cyclic stress, another way to induce cell cyclic stress and a measurement of the cyclic straining as a readout are mandatory. Figure 1 results intend to address the relation of cyclic stained cells with its viability. But the in vitro device used by the authors is only shown in figure A without a clear proof that cyclic stress was induced to the cells; The mechanical stimulus must be measured and not limited to observe and count live/dead cells by its application. I found in summary the results presented in figures 1 and 2 not sufficient to objectively access cell viability in such artificial conditions, without a measurement of the biophysical stimulus to create mechanical stress to the cells, or to show by comparison with distinct types of mechanical stress. If other researchers want to replicate your cyclic straining to their biological systems, it will be impossible to do it with the descriptions and assays you provide in this manuscript.
The section 2.3 -cell morphology an FAs should start after figures 3 an 4, and some of the information in these to figures can be considered to be added as supplementary material, since there is no need to present in the main section of the manuscript plots of absorbance with the treatment with distinct drugs with or without mechanical stimulus being with no significant difference and no description of what absorbance at 450nm is giving, since I believe from the scarce description that the EdU incorporation was measured by counting positive cells in each condition. What the plots of Absorbance at 450 nm are assessing?
Its good however that now the authors included a human melanoma cell line together with the results first demonstrated with B16 cells. In fact, these results using A375 melanoma cells should be taken more in consideration since a clear increase with MS was observed in the percentage of EdU incorporation (double in the case of A375 cells, whereas in B16 cells there was an increment of only 10%.
In line 132, the authors state that figures 4 and 5 show cell spreading an elongation. This is not true with the addition now of A375 results, and figure 4 is no longer assessing cell spreading and elongation. Also, the correlation made by the authors of cell elongated morphology with cell proliferation is to speculative.
Concerning the FA’s size quantification, the results presented in plots (figures 5 and 6 D) for the total area of FAs is not corroborated with the representative images shown. In fact, in case of B16 cells, by the images in figure 5A it would be expected a decrease in FAs. From the material and methods section (line 258), it is not clear how the authors quantified the FAs area? Which formula was used for this quantification based on the focal adhesions of the fluorescent images?
Finally, the suggestion that the regulatory expression of melanoma specific genes may relate to TFs through mechanical transduction is in my opinion to subjective (Line 190) and not corroborated with results presented by the authors.
There is no obvious conclusion in the manuscript besides the speculation made by the authors in the final lines of the abstract.
In my opinion, the manuscript still needs to be highly improved with results that corroborate the statements and more organized to drive a clear message before it is ready for publication.
Reviewer 2 Report
The authors have addressed my questions. It would be valuable to make the RNAseq data accessible to the scientific community by depositing them either in the GEO database or in the NCBI's SRA.
Author Response
Thank you very much for your kind reviewing. The original RNAseq data will be uploaded to GEO database latter.
Round 3
Reviewer 1 Report
Dear authors,
I believe that has been an effort to continously improve this manuscritp.
I have one more suggestion to make, concerning the conclusions of the manuscript; it would be relatively easy and improve your work if a schematics or working model of your findings could be made based on section 4, conclusions.
additionaly, I believe in line 203 the authors want to mention cell spreading and elongation and not well spreading and elongation.
Author Response
Thank you very much for your kind and careful reviewing.
The schema of mechanism was added and the sentence was revised in the revised manuscript.